# The State-of-the-Art of *Mycobacterium chimaera* Infections and the Causal Link with Health Settings: A Systematic Review

**DOI:** 10.3390/healthcare12171788

**Published:** 2024-09-06

**Authors:** Vittorio Bolcato, Matteo Bassetti, Giuseppe Basile, Luca Bianco Prevot, Giuseppe Speziale, Elena Tremoli, Francesco Maffessanti, Livio Pietro Tronconi

**Affiliations:** 1Astolfi Associates Legal Firm, 20122 Milan, Italy; 2Department of Health Sciences (DISSAL), University of Genova, 16132 Genova, Italy; 3Infectious Diseases Unit, IRCCS Ospedale Policlinico San Martino, 16132 Genoa, Italy; 4IRCCS Orthopaedic Institute Galeazzi, 20157 Milan, Italy; basiletraumaforense@gmail.com (G.B.); luca.bianco96@gmail.com (L.B.P.); 5Section of Legal and Forensic Medicine Clinical Institute San Siro, 20148 Milan, Italy; 6Residency Program in Orthopaedics and Traumatology, University of Milan, 20122 Milan, Italy; 7GVM Care and Research, Anthea Hospital, 70124 Bari, Italy; gspeziale@gvmnet.it; 8GVM Care and Research, Maria Cecilia Hospital, 49033 Cotignola, Italy; etremoli@gvmnet.it (E.T.); fmaffessanti@gvmnet.it (F.M.); ltronconi@gvmnet.it (L.P.T.); 9Department of Human Science, European University of Rome, 00163 Rome, Italy

**Keywords:** healthcare-associated infections, *Mycobacterium chimaera*, cardiac surgery, causal link

## Abstract

(1) Background. A definition of healthcare-associated infections is essential also for the attribution of the restorative burden to healthcare facilities in case of harm and for clinical risk management strategies. Regarding *M. chimaera* infections, there remains several issues on the ecosystem and pathogenesis. We aim to review the scientific evidence on *M. chimaera* beyond cardiac surgery, and thus discuss its relationship with healthcare facilities. (2) Methods. A systematic review was conducted on PubMed and Web of Science on 7 May 2024 according to PRISMA 2020 guidelines for reporting systematic reviews, including databases searches with the keyword “*Mycobacterium chimaera*”. Article screening was conducted by tree authors independently. The criterion for inclusion was cases that were not, or were improperly, consistent with the in-situ deposition of aerosolised *M. chimaera*. (3) Results. The search yielded 290 eligible articles. After screening, 34 articles (377 patients) were included. In five articles, patients had undergone cardiac surgery and showed musculoskeletal involvement or disseminated infection without cardiac manifestations. In 11 articles, respiratory specimen reanalyses showed *M. chimaera*. Moreover, 10 articles reported lung involvement, 1 reported meninges involvement, 1 reported skin involvement, 1 reported kidney involvement after transplantation, 1 reported tendon involvement, and 1 reported the involvement of a central venous catheter; 3 articles reported disseminated cases with one concomitant spinal osteomyelitis. (4) Conclusions. The scarce data on environmental prevalence, the recent studies on *M. chimaera* ecology, and the medicalised sample selection bias, as well as the infrequent use of robust ascertainment of sub-species, need to be weighed up. The in-house aerosolization, inhalation, and haematogenous spread deserve experimental study, as *M. chimaera* cardiac localisation could depend to transient bacteraemia. Each case deserves specific ascertainment before tracing back to the facility, even if *M. chimaera* represents a core area for healthcare facilities within a framework of infection prevention and control policies.

## 1. Introduction

A case report by Vendramin et al. describes the current upper extreme of *Mycobacterium chimaera* (*M. chimaera*) infection latency in a subject who underwent cardiac surgery 12 years earlier [1]. This interesting case poses the opportunity for an in-depth literature review oriented towards a causal assessment between *M. chimaera* infection and healthcare settings. The issue of defining an infection acquired in a healthcare setting, a so-called healthcare-associated infection (HAI), is of central significance from a medico-legal point of view. In the case of patient harm, in fact, it could derive the attribution of the restorative burden to the healthcare facility, and there is a growing interest in the implications in terms of the reliability and risk management strategies in healthcare systems [2,3,4,5,6,7]. The article mentioned above reported on *M. chimaera* infection from a clinical perspective and placed it in association with a previous cardiac surgery of an aortic dissection repair with graft replacement of the ascending aorta, based on a chronological and topographical criterion. But the farther apart two natural events are, the more other events can occur in between, and the more complex the causal reconstruction becomes [8]. The referenced scientific literature on *M. chimaera* is precisely limited to the *M. chimaera* outbreaks related to a lot of contaminated heater-cooler units, starting with the first case reported in the USA in 2013 and distinct cases in Italy between 2016 and 2019 [9,10,11,12]. That extreme case, however, does not describe any genetic ascertainment to be traced back at least to the group of strains most frequently identified in contaminated devices or in strains contaminating healthcare facilities [13,14,15,16,17,18]. And the same Italian Health Ministry Recommendations from 9 January 2019, starting from the European Centre for Disease Prevention and Control (ECDC) protocol and epidemiological criteria, refer to the temporal criterion: ‘performance of surgery that required cardiopulmonary bypass in the six years preceding the onset of symptoms of infection’ [19,20]. Nevertheless, this definition responds to epidemiological and clinical necessity, being biassed towards sensitivity. It was appropriately stated that *M. chimaera* infection poses significant difficulties for various reasons: (a) the time between infection and the manifestation of symptoms is extremely long and are not yet fully understood or described in the literature; (b) the characteristics of this particular mycobacterium and the route of infection are not widely recognised; (c) many diagnostic tests for *M. chimaera* are slow and of low sensitivity; (d) the healthcare infection route is very specific and occurs through routine processes employed within operating rooms; and (e) the elimination of *M. chimaera* biofilms from surfaces using routine disinfection procedures, both in healthcare settings and in communities, is challenging [21]. These considerations depend, and it is worth pointing this out, on the related scientific knowledge that, outside the perimeter of complications after cardiac surgery [22], still displays several unsolved challenges. Moreover, we are confronted with very recent microbiological acquisitions in terms of identification and pathogenic characteristics, being already relevant to the re-evaluation of cases previously attributed to Mycobacterium intracellular, or others from the avium complex group, and then traced back to chimaera [23,24]. Vendramin et al. themselves, indeed, emphasise the importance of punctual microbiological, histo-pathological, pathological-forensic, and genetic investigations, as well as the reporting and reviewing of cases with atypical clinical presentations during the follow-ups of patients who have undergone cardiac surgery [1]. The aim is precisely because it is only from robust and reliable data that is it possible to infer a clear causal relationship.

Therefore, the purpose of this systematic literature review, particularly in the face of such long latencies between healthcare interactions and clinical manifestations, is to summarise the scientific evidence on *M. chimaera*, which is not only restricted to cardiac surgery, to assess the possibility of alternative causes of transmission and infection besides the hypothesis of in situ deposition of aerosolised particles, and then to evaluate the role of healthcare settings.

## 2. Materials and Methods

A systematic literature review was conducted according to the PRISMA 2020 guidelines for reporting systematic reviews, including searches of databases and registers only [25]. The search was conducted on PubMed (PubMed.gov; US National Library of Medicine National Institute of Health) and Web of Science Core Collection (Clarivate) using the keyword “*Mycobacterium chimaera*” without any filter, any Boolean operator or limitation. PubMed query detail is “*Mycobacterium chimaera*” [Supplementary Concept] OR “*Mycobacterium chimaera*” [All Fields] OR “*Mycobacterium chimaera*” [All Fields]. Web of Science Core collection query detail is ALL = (MYCOBACTERIUM CHIMAERA). The literature survey was performed on 7 May 2024. The PRISMA 2020 flow diagram for systematic reviews was used for article screening, together with PRISMA checklist (Appendix A Appendix A). PROSPERO does not accept scoping reviews, literature reviews or mapping reviews. Not applicable. Duplicate articles were semi-automatically removed before screening by the first author (VB). Then, the articles were screened by the authors (VB, MB, LBP), who reviewed the title and abstract and excluded all not relevant articles, such as those incidentally referring to M. chimaera without any data on the pathogen, incomplete articles, meeting abstracts, and not available articles.

Articles deemed eligible for review were then independently re-screened by three authors (VB, MB, LBP) according to the following inclusion criteria: cases not or improperly consistent with in situ deposition of aerosolised *M. chimaera* particles during cardiac surgery, such as clinical manifestations not in the cardiac tissue/district after cardiac surgery or atypical clinical manifestations after other surgical intervention not using HCU, or in general cases of *M. chimaera* infections. Divergences were solved through discussion.

Included articles were discussed among all authors and summed in a table. The following items were considered: number of patients per each study, surgery if any, time lapse to presentation in case of surgery, Country, organ/tissue involved with complete diagnosis. The setting was reported distinguishing between not healthcare related (prevalent daily life, environmental contact and signs/symptoms presentation in community setting) or healthcare related because of frequent hospitalisation and/or pathologies requiring hospitalisation (underwent to surgery, possible frequent healthcare contact, possible healthcare contact).

The risk of bias assessment for the included articles was not applicable, as it was descriptive research. Articles reporting clinical cases that met the inclusion criteria were considered valid because they answer the study aim (qualitative data present or absent). The Limitations section contains further considerations.

*M. chimaera* characteristics and ecosystem, health devices and implants contamination, incubation period and clinical presentation, lung presence, transmission modality, detection and disinfection techniques were then discussed, comparing them with the findings of the considered articles, to finally argue on causal link with healthcare setting.

## 3. Results

Overall, the search yielded 256 articles on PubMed and 323 on Web of Science, reduced to 318 after removal of duplicates (Figure 1).

After the first screening, 28 articles were excluded because not relevant or non-retrievable, resulting in 290 distinct articles, including 26 review and 5 systematic reviews.

Among them, the most recent systematic review on *M. chimaera* infection after cardiac surgery was performed by Wetzstein et al., set at 15 June 2022, reporting 54 articles for a total of 180 cardiac surgery patients worldwide. The median period between the time of surgery and the occurrence of the first symptoms was 17 months (interquartile range 13–26 months), with 80% of the patients showing symptoms within 2 years. After that review, 49 articles were published on *M. chimaera* in general, without limitations to cardiac surgery. Eight articles (33 patients) were linked with cardiac surgery. Overall, to date, 62 articles (213 cardiac surgery patients) worldwide describing *M. chimaera* infection consistent with the hypothesis of in situ deposition after cardiac surgery were reported. The median incubation time in the most recent articles was in line with previously reported data.

The 290 eligible articles were deemed eligible and re-screened for inclusion. An established diagnosis of *M. chimaera* infection without clinical manifestations in the cardiac tissue/district after cardiac surgery or atypical clinical manifestations after other surgical intervention not using HCU or involvement of other organs or tissues not consistent with the hypothesis of in situ deposition of aerosolised *M. chimaera* particles, emerged in 34 articles (377 patients) (Table 1).

Out of these, in 5 articles, 12 patients had undergone cardiac surgery: 2 presenting symptomatology to be referred to the musculoskeletal district with a diagnosis of spinal osteomyelitis and discitis, 1 with a diagnosis of hand tenosynovitis with history of ipsilateral elbow wound, 4 with a disseminated presentation without endocarditis, and 5 patients with a disseminated presentation without endocarditis and hepatitis-like signs. Moreover, 1 patient showed urinary district *M. chimaera* infection after kidney transplantation with unknown infection source, concomitant *Mycobacterium malmoense* lung infection and lung fibro anthracosis.

The remaining 28 articles (364 patients) reported *M. chimaera* infections without evidence of cardiac signs or symptoms and/or without relation with any surgical approach or presence in operating room: 10 articles (10 patients) reported lung related signs and symptoms usually with various concomitant pathologies; of note, one of this article described the case of a healthcare worker reporting lung involvement, ten years after working as operating room nurse; 11 articles (346 patients) reported *M. chimaera* diagnosis after respiratory specimen reanalysis; in particular, 1 article reported data from genomic analysis of respiratory specimens in patients with cystic fibrosis to define cluster correlation to environmental samples in hospital outbreaks.

Three articles (3 patients) described disseminated infection in immunocompromised patients (2) and in presence of osteomyelitis (1). A single article (2 patients) described meningeal involvement. Finally, 1 patient showed hand tenosynovitis in rheumatoid arthritis, 1 patient a skin ulcer on the face (chin) with concomitant oral cavity periapical abscess, and 1 patient central venous catheter infection with recurrent *M. chimaera* blood and sputum positivity.

## 4. Discussion

### 4.1. Mycobacterium chimaera’s Characteristics and Ecosystem

*Mycobacterium chimaera* is a non-tuberculous mycobacterium (NTM), first described by Tortoli et al. in 2004, that belongs to the Mycobacterium avium complex (MAC). It is an opportunistic human pathogen, ubiquitous in the environment and water-borne, typically found in groundwater or tap water, soil, house dust, domestic and wild animals, and birds [59,60,61,62]. However, its prevalence in the environment is mostly unknown [9,63,64,65]. It is a slow-growing, non-pigmented, acid-fast positive, distinguished by non-motile and non-spore forming coccobacilli with a growth that takes up to 6–8 weeks [9]. The highly lipophilic cell wall, the low number of porins associated with a variety of efflux pumps, inducible resistance mechanisms and biofilm production confer an environmental resistance and a natural drug-resistance as well, similarly to other species belonging to the MAC group. The resistance against frequently used disinfectants, such as chlorines and ozone, could also explain their persistence in water systems. Indeed, water stagnation and high temperature, up to 40 °C, promote the formation of biofilms and create a more favourable environment for *M. chimaera*, while its growth is considerably slow at temperatures in the range of 25–35 °C [17,18]. In addition, MAC members and *M. chimaera* preferentially colonize warm water sources, and this might, at least partially, explain their propensity to aerosolize [9,63,66]. Moreover, the presence of NTM in dust collected from residences clearly suggests that some NTM species are desiccation tolerant, resulting in a further complexity, not restricting the survival to humid ecosystems [67]. Differences in water treatment, water heating systems, source water type, showerhead design, and characteristics of the water distribution systems, determine a lower presence of mycobacteria and *M. chimaera* in Europe than in the United States. In fact, U.S. households’ water had significantly higher chlorine and iron concentrations, but lower pH and nitrate levels than European households on municipal water [68]. Previous studies reported the presence of NTM and *M. chimaera* in domestic drinking water, showerheads and washing machines, with clear association with lung diseases and air diffusion [69,70,71,72]. Durnez et al. has demonstrated the presence of *M. chimaera* in small mammals in urban areas, highlighting their potential role as carriers of the pathogen [65].

### 4.2. Heater-Cooler Units, Medical Devices, Water, and Air-Conditioned Implants

An outbreak of *M. chimaera* infection following cardiac surgery has been reported, and the use of a specific heater-cooler units (HCU) model, clinically available since 2006 and contaminated in production, has been identified as the potential source. However, contamination of other HCU brands and hospital water systems has been observed and that suggests that the infectious risk might persist even after carefully controlling the single lot of devices contaminated in the production line [14]. The primary mode of transmission has been identified in the contaminated water tanks of HCUs, thermoregulatory components of Extra-Corporeal Membrane Oxygenation (ECMO), near the rotating fan, which can produce a contaminated aerosol within the operating room. The ECMO machines are air-tight and closed systems, unlike the HCUs used in cardiothoracic surgery, which may have precluded the release of infectious aerosols, in case of contamination, by those devices [73]. *M. chimaera* contamination of ECMO devices was reported by Trudzinki et al., but without the presence of air samples or other environmental samples in the surgery room. Nonetheless, transmission of *M. chimaera* from an ECMO device, in case of circuits damage during use, has been reported reported, due to haematic line failure, and this widens the problem to other devices sharing similar characteristics and use [74]. In fact, similar risks have been reported for infusion heating devices [75]. Also, as a consequence of Health Ministries’ campaigns held worldwide, which recommended the reporting of cases clinically attributed to the pathogen in the follow-up of patients undergone cardiac surgery, attention was focused in the first instance on the structural architecture of HCUs: some models, due to the proximity of fans and water tanks, have an easier and higher possibility to aerolize pathogens’ particles. Furthermore, their location in the operating theatre and the laminar air flow [76,77,78], the supply of water tanks [79], healthcare facilities’ water contamination and healthcare hand washing machines [80,81,82], overall operation time, the use of other medical equipment such endoscope reprocessing or haemodialysis systems, were reported as influencing factors that could predispose to *M. chimaera* contamination [14,75,83,84]. All these factors gradually extended the issue to the management of aeraulic systems in healthcare facilities, with the associated sampling. Delving into the subject, it was thus observed that mycobacterium could be almost widespread in the healthcare setting. Therefore, leaving aside for a moment whether an alternative transmission to the direct contamination of the surgical site is possible, apart from aerosolised particles, the healthcare facilities can be contaminated by *M. chimaera* and overall the so-called opportunistic premise plumbing pathogens (OPPP) depending on the replication of those natural ecosystems with certain ecological niches of choice, particularly warm water, where then can widely produce biofilms [63,85].

### 4.3. Incubation Period and Symptoms Presentation

The literature showed an extremely variable latency time, with extremes ranging between 6 weeks and 12 years, although 80% of the cases manifest symptoms within two years [11,86]. The clinical spectrum comprehends prosthetic valve endocarditis, grafts and prosthetic-related parts infection, sternotomy wound infection, pleuro-pericarditis and mediastinitis, usually with disseminated extrapulmonary infection. *M. chimaera* shows an ease of systemic dissemination, which results in a dismal prognosis, as only a 3-year survival from diagnosis of 40% has been observed. Embolic and immunologic sequela of disseminated infections have also been reported, such as splenomegaly, arthritis, spinal osteomyelitis, cytopenia due to bone marrow involvement, even without cardiac tissue involvement [11]. Specifically, on *M. chimaera* infective endocarditis, no natural cases have been described in the literature except ‘after’ cardiac surgery related to the outbreaks, but it must be considered that *M. chimaera* is a very recently defined pathogen among NMT which has only recently been included in the endocarditis diagnosis flow-chart, while even rarely other mycobacterial endocarditis are described [11]. For instance, a review has recently identified 20 cases reporting infective endocarditis caused by *Mycobacterium abscessus* [87]. Moreover, a wide range of infective endocarditis remains with a negative blood culture (i.e., with an undefined pathogen) with rates ranging from 7.7% to 66% (mean 10–20%). The variability in the observed incidence can be explained by several factors, most notably the lack of specific clinical syndromes related to *M. chimaera* infections [88,89], local variations in the early use of antibiotic therapy prior to obtaining blood cultures and the fact that mycobacterial cultures are not routinely pursued in clinical setting, but also differences in testing strategies and geographic variation of specific organisms [90,91]. Of course, it should also be considered how the same diagnostic challenge leads to an alteration of the natural history of the disease, for instance through empirical therapies. The same forensic investigations are not always feasible and/or possible within a reasonable—and so significant—timeframe for pathogens research, while histopathological assessment of removed prosthetic tissues must always be pursued.

### 4.4. Presence in the Lung System

Several article and patients reported lung system involvement, usually associated with cystic fibrosis, obstructive lung disease or pneumoconiosis, or, more in general, immunocompromising conditions. Pulmonary infection identical to that of other MAC infections have been reported [23]. Patients may suffer from chronic cough and breathlessness, with or without associated constitutional features such as fatigue, fever, night sweats and weight loss. Chest imaging may reveal nodules and hilar lymphadenopathy, and several patients have been erroneously diagnosed with sarcoidosis or pulmonary tuberculosis before the diagnosis of *M. chimaera* infection was made [92]. Mc can also cause pulmonary infection, like *M. avium* and *M*. *intracellulare* [23,33,42,93]. In the respiratory tract, Mc resulted more likely to be a colonizer and less likely to cause true infection, compared to the other MAC species, yet enhancing the difficulty of the prevalence assessment in the community. In fact, patients with symptomatic respiratory infection due to Mc were more likely to be immunosuppressed, suggesting an opportunistic pattern of the pathogen, aligned with the reported low virulence [17]. The prevalence of Mc has also been addressed in studies assessing whether the reservoir for *M*. *intracellulare* pulmonary disease was the patient’s household water system [17,33,94]. In a German cohort, Schweickert and colleagues, after genetic sequencing on 166 isolates that had been previously classified as *M*. *intracellulare*, found 86% (143/166) of these samples to be *M. chimaera* [95]. The discrepancy between Tortoli et al. high virulence and Schweickert et al. may be the result of a sampling bias, with patients suffering from a more severe disease in the former, as all their patients were hospitalised and underwent cardio-surgery. Low virulence, together with the indolent symptomatology and the ease of systemic dissemination resulted in an unfavourable prognosis. Numerous studies have commented on environmental exposures to water supplies, household and environmental, as a risk factor for MAC pulmonary infections [96].

### 4.5. Modality of Transmission

Most recent cases referring to *M. chimaera* infection, support a relation to the contamination of a heat exchanger by experimental investigations of the contamination of the machine’s water tanks, a water temperature favourable to proliferation, an air flow that favoured aerosolization of the pathogen from the water and its dispersion in the air of the operating theatre [97]. The aerosolised pathogen subsequently adheres to the exposed substrates causing mediastinitis, infection of the surgical site, endocarditis, pleuritis; the immune response capacity and the formation of biofilm on an inert substrate were other relevant predisposing factors. Interestingly, it is not a valve that is contaminated ab initio, but a valve, tissue, graft, device that has been contaminated in a later stage, as mentioned most often as a function of aerosolised particles released by HCUs during laparotomic operations with cardio-pulmonary bypass [11,98,99]. This is supported by the presence in the air in case of active devices, the reduction of the presence when the device is switched off, the orientation of the airflow on the operating table, as well as the association with longer operations [10,89,100]. Yet, patients showed higher risk of *M. chimaera* infection when the ECMO used for open surgery, than for respiratory failure, posing the role of the larger potential entry sites for the pathogen [9,83,89]. Literature has also described the similarity of the genetic trait of infection samples with those present in the machinery such as HCU, thus supporting a more stringent relationship between the use of machinery in healthcare and therefore assessing HAI definition [13,101]. Nevertheless, as said considering its distribution in the ecosystem and lung colonization/infection, the importance of this pathogen to human health is beyond the cardio-thoracic surgery related outbreak, as in general for the mycobacterium family. Our knowledge on this emerging pathogen relating population prevalence, mode of infection and virulence mechanism is still limited [102]. We reported evidence of complicated diagnoses that does not meet or meet improperly the topographical criterion with respect to aerosolization of the pathogen in the operating room and the subsequent direct deposition. In particular, Chand et al. reported three cases of late disseminated *M. chimaera* infection and one osteomyelitis after valvular surgery without endocarditis and/or other infections directly matching with *M. chimaera* aerosol deposition, additionally demonstrating high genetic similarity of the samples to HCU isolates [36]. These few cases seem consistent with an in-hospital source, but the presence in the blood stream at such long intervals needs still need an explanation, as it may be secondary to a subsequent care-related cause as well as an undiagnosed, but apparently non cardiac site, of dormant *M. chimaera*. As mentioned, although the modes of transmission have been poorly investigated, the presence, persistence and growth in household and outdoor plumbing, account for the need to weigh the risk between healthcare and community contexts [17,68,94]. Certainly, the presence of high-risk subjects in such care settings favours their detection, but the precise association and definition of risk must necessarily depend on a more precise analysis of exposure in community environmental subsystems, enlarging the sample to include non-hospitalised/medicalised subjects, thus avoiding any potential selection bias [33,103]. Lung infections and lung colonisations were found in non-hospital settings, supporting the hypothesis of community aerosolization [33], even if the healthcare worker with lung involvement could be linked with the contagion in operating room, even though this activity goes back 10 years, and no genetic assessment has been carried out [40]. In other words, just as for HCUs after outbreaks, because of atypical clinical pictures in patients undergoing major cardiac surgery, attention has been focused on a little-known mechanism of exposure and transmission accidentally occurring, so it will be necessary to ascertain the presence, exposure and mode of transmission in another context and in an immunologically healthy population [33,103]. The localisation in the choroid, in spinal vertebrae in the form of a discitis, in the bone marrow in the form of granulomas, in chronic wound ulcers, need to be explained according to aerosol contamination, rather than supporting the hypotheses of localization by a circulating pathogen or not health-related contagion [50]. Interestingly, Asadi et al. reported and discussed a case without valvular involvement and without abscess formation, but with the presence of colonization of the peri-graft tissue at histopathological examination. Similarly, paravalvular or peri-graft involvements as disseminated infection without valve endocarditis were reported by Kohler, Clemente and Natanti, Trauth and Shaeffer. Those authors stressed the need for histopathological examination, as further confirmation, and once again the role of haematogenous spread, granulomas formation, and the struggle of endocarditis diagnostic path [92,104,105,106,107]. The disseminated forms themselves support pathogen circulation and haematogenous spread, which is shared by the entire mycobacterial family, as well as with mechanisms of embolization and dissemination via the meningeal route [45,108]. Most important in explaining modes of transmission and possible community acquisition is the finding of *M. chimaera* infection in non-hospitalised individuals or in surgeries that do not use HCUs. To date, this systematic literature review reported rare data about systemic manifestations and tissues involvements not directly ascribable to a topographical criterion which complies with the health-related aerosolised pathogen’s deposition. Alternative transmission routes are however possible and described. Patient blood may have been contaminated by a leakage in the membrane in one of the heat exchange units, but this should be regarded as an exceptional event; alternatively, contamination or colonization of the prosthetic material due to intravenous catheter-related bacteraemia may be considered an alternative explanation [22]. In addition, lung and disseminated infections are possible outside the operating room, such as by inhalation of aerosols or ingestion of contaminated water in various setting, for instance from community showerheads [66]. While these cases have certainly been described less frequently, limited to single case reporting, they showed atypical cases that could also be explained by aerosol inhalation and subsequent spreading via bloodstream [22,59].

### 4.6. Detection

Detection of *M. chimaera* relies on multiple methods, including standard microscopy, proteomics, molecular assays and next-generation sequencing, with different advantages and pitfalls, both for clinical and environmental samples, as well as the microbiological basis of the process of linking a case of *M. chimaera* infection to an environmental exposure, partially as for other opportunistic premise plumbing pathogens [19,73,109,110]. The algorithm for sample processing and staining for clinical suspected *M. chimaera* infection follows the same techniques and principles in place for other NTM species, according to each laboratory standard operating procedure involving decontamination and staining with auramine fluorochrome stain, carbol fuchsin-based Ziehl-Neelsen or Kinyoun stains for direct microscopy in suitable samples. Of note, clinical samples for *M. chimaera* infection investigation can range from respiratory specimens, urine, hepatic biopsies, bone/soft tissue biopsies to prosthetic material/graft tissues sampling. The identification of *M. chimaera* isolates from clinical/environmental samples followed ECDC protocol released in 2015, aiming to the harmonization of samples and data collection in cardiothoracic setting, and the International Society of Cardiovascular Infectious Diseases Guidelines [19,111]. Those investigations were graphically summarized by Cannas et al. in 2023 [100]. Concentration of water samples can be performed with filtration or centrifugation methods; for *M. chimaera*, filtration has proven more sensitive than centrifugation. After concentration, water samples should undergo decontamination, preferably with cetylpyridinium chloride, which has proved superior to the more common standard solution of N-acetyl-L-cysteine sodium hydroxide (NALC-NaOH) or NaOH in recovering MAC from environmental samples. Air samples should be obtained with an environmental air impaction sampling device capable of sampling a certain volume of air in each period. Environmental samples are than directly inoculated into selective liquid and solid media for mycobacterial culture. Culturing and further characterization requires a dedicated level of expertise and laboratory resources which are usually located in mycobacteriology units processing *M. tuberculosis* complex samples. For NTMs in general, performance of both liquid and solid media is recommended for clinical samples. Using both media has shown to increase the sensitivity of NTM detection by 15%, although the choice on the preferred type of solid media (Middlebrook 7H11 or 7H10 or Löwenstein-Jensen agar) is less evident in terms of performance. As the other species of the MAC complex, *M. chimaera* optimally grows with temperature ranging in the 36–42 °C interval. For environmental sampling, in case of limited resources, solid media is the preferred option according to the ECDC protocol, although some evidences point to a greater sensitivity of MGIT in detecting *M. chimaera* from water samples [112]. As *M. chimaera* grows in 6–8 weeks, incubation at 35–37 °C for 8 weeks is recommended with weekly inspection for mycobacterial growth and colony counting.

Also, identification of *M. chimaera* is challenging and should be performed in reference laboratories equipped for molecular diagnostics of mycobacteria. Guidelines for species identification follow the general guidelines for NTM laboratories and include molecular -preferred- and proteomics techniques, although the latter do not perform optimally for *M. chimaera* identification. Nucleic acid sequencing of DNA regions among internal transcribed spacer (ITS), 16S rRNA, rpoB and hsp65 is currently the reference method to identify *M. chimaera* [113]. Another approach involves the use of commercial line probe assay INNO-LiPA Mycobacteria, which carries a specific probe -the internal transcribed spacer between gene 16S e 23S of the rRNA—to distinguish species among Mycobacteria [19]. Indeed, several commercial easy-to-use kits are available for mycobacterial identification, but their performance in identifying *M. chimaera* is variable [100]. A study comparing three molecular biology kits (INNO-LiPA Mycobacteria, GenoType bacterium CM and GenoType NTM-DR) and matrix-assisted desorption ionization–time of flight mass spectrometry (MALDI-ToF MS) against reference standard in *M. chimaera* identification found a high level of concordance of INNO-LiPA Mycobacteria and GenoType NTM-DR, while GenoType Mycobacterium CM and MALDI-ToF MS were not able to distinguish between *M. chimaera* and M. intracellulare, which are genetically similar [114]. Recently, high sensibility and sensitivity are reported for NTM detection with MALDI-ToF MS [115]. Whole-genome sequencing (WGS) is currently the gold standard for performing genomic typing and has been pivotal during the *M. chimaera* outbreak to track cases and to link clinical cases to environmental exposure. The ECDC supports its use in genetic analysis of *M. chimaera* by sharing sequences of isolates among laboratories and countries to help informing about the genomic diversity of *M. chimaera* across Europe. Studies on the global outbreak have shown that WGS can be used to link clinical samples to HCUs contamination, also describing phylogenetic tree and the more prevalent branches involved [14,101,116].

### 4.7. Disinfection

Often HCUs do not provide any technology to reduce bacterial or other contamination and maintenance instructions for the HCU highlight the risk of oxygenator damage and of heat exchanger leakage with some corrosive disinfectant agents, with subsequent impair of the heat exchanger permeability and function. Changes in the overall structures of HCU models resulted efficacious in reducing at least the risk of aerosolization [117]. Disinfecting HCU is then very difficult [118]. Chemical agents or the combination of chemicals such as chlore derivative, chlorhexidine-alcohol, peracetic acid and sodium hydroxide [119,120,121,122] were reported, with partial or temporaneous efficacy or limited to colony forming units load reduction [123,124]. With water chlorination the paradoxical effect of increasing *M. chimaera* presence through amebae symbiosis was also reported [125]. The same water chlorination impact on *M. chimaera* presence in water was reported by Virdi et al. in U.S. and E.U. [94]. The European Centre for Disease Prevention and Control recommended the use of hydrogen peroxide in filtered water to fill heater-cooler unit tanks; however, heater-cooler units became heavily contaminated by opportunistic waterborne pathogens such as Pseudomonas aeruginosa and *Stenotrophomonas maltophilia* [117,121]. Silver-ion cleaning also resulted insufficient, considering mainly resistant and structured biofilm formation [126]. As an alternative, or in support to chemical and thermal disinfection, filtration method resulted effective in the reduction of pathogen load [120,127]. Filters and UV disinfection have been evaluated for their ability to reduce numbers of waterborne non-NTM organisms from drinking water, but their efficacy in reducing NTM counts is not well-established [64]. Pradal et al. demonstrated *Methylobacterium* sp. extract efficacy in reducing biofilm of *M. chimaera* strain, while Foltan et al. suggested the use of electrolysis [128,129].

### 4.8. Causal Link Assessment

*M. chimaera* infection presents peculiar characteristics, common to the so-called opportunistic premise plumbing pathogens among which Legionella pneumophila. However, regarding *M. chimaera*, the overall environmental prevalence and transmission modality, together with the haematogenous spread, still deserve in-depth experimental studies to assess the possibility of tissue localisations starting from bacteraemia and further abscess formation [47,85,97]. For our purposes, i.e., the causal tracing of the infection back to the healthcare context, the data presented are useful to emphasise how even more with long latency periods, without long hospitalisations, and thus of main permanence in the living/social context, the probability of the subject being exposed to the pathogen has no specific or absolute prevalence. Therefore, a domestic contraction cannot be excluded a priori, and recent personal history must be investigated, as regarding Legionella pneumophila infections and hospital outbreaks. Senescent and poorly maintained water supply systems, poor environmental hygiene, extreme humidity or standing water, extreme branches of the water coiling system are significant factors also in residential settings, but even recent air travels could be relevant, as for Legionella [130,131,132]. Furthermore, the insufficiency of data on environmental and domestic prevalence, the recent studies on its ecology, the selection bias of the medicalised sample, as well as the infrequent use of robust ascertainment such as genetic and histopathological one, and not least the diagnostic complexity of endocarditis in general, severely limiting confirmation in developing countries, are further elements to be weighed up [104,133]. The same topographical criterion, i.e., the presence of contamination, up to abscess forms, at the site of previous surgery (valve replacements, vascular prostheses, suture, graft tissues, peri-prosthetic tissues and materials) must also be studied with respect to the presence of a foreign bodies interposed to the systemic flow in case of bacteraemia. This can easily be a colonisation site, escaping immunological control, and with the possibility of proliferating in a subtle and slow manner. This is even more than a possibility if one considers that the *M. chimaera* exploits the formation of biofilms and proliferates where the immune component is less competent and effective. As the phenomenon is frequently associated with the peculiarities of the healthcare setting and its contamination, due to the limitation of hospitalised population, each case must be studied individually and in detail, before tracing the cause directly to the facility. In fact, a hypothetical relationship between two naturalistic events could be the basis for a diagnostic hypothesis and then for an empirical therapy, but this cannot be the diagnosis from a law system perspective and for the purposes of possible compensation for health damages, since a tighter causal ascertainment is required. And as for endocarditis, even for early endocarditis occurring in the first year, the definition as healthcare-associated cannot be the same as other bacterial infections, as it does not matter that this occurs after having been hospitalised for more than 48 h, also from an epidemiological point of view [134]. Clinical data therefore must be integrated with the remaining criteria for causal ascertainment, namely the qualitative-quantitative suitability and the logical-scientific probability [135].

In the technical evaluation of each single case, the fact that contagion from *M. chimaera* is possible need to be in-depth discussed and the specific source of infection found out in that hospital, due to the use of contaminated HCUs or to other types of water/device contamination and having taken care to assess genotypic correlation with the healthcare fomites and finally exclude other possible causes. Hence the causal relationship with the healthcare setting and hence the definition of healthcare associated infection.

## 5. Limitations

Although the searches in PubMed and Web of Science were rigorously and widely conducted, it is possible that some cases that met the inclusion criteria were not found. However, the cross-reference analysis did not reveal any additional articles reporting clinical cases that were not included in this review. Publication bias must also be considered, as single or atypical clinical presentations, unrelated to cardiac surgery or surgery in general, may not have been described in the literature or only reported at conferences. This could be due to a greater focus on the association with cardiac surgery. In addition, the systematic review conducted is descriptive by nature, so it is not possible to determine the robustness of the article included, while the medico-legal perspective accounts for the wide heterogeneity of the included clinical case and overall cited literature. Independent and multidisciplinary screening and review of the literature could have partially weighed those limitations.

## 6. Conclusions

*Mycobacterium chimaera* infection is a recent scientific acquisition and presents peculiar characteristics, such as the ecosystem of proliferation, resistance strategies, transmission modality, high mortality and the initial relationship with a healthcare context that remains not fully elucidated. Further studies on the in-house water system prevalence and the hypothesis of haematogenous spread and rooting on the valve site are needed. Nonetheless, healthcare contamination represents a core and dedicated area for healthcare facilities within the framework of infection prevention and control policies and overall hospital ecosystem monitoring, looking methodically and with periodical monitoring at aeraulic implants, medical devices and surfaces hygiene.

## Figures and Tables

**Figure 1 healthcare-12-01788-f001:**
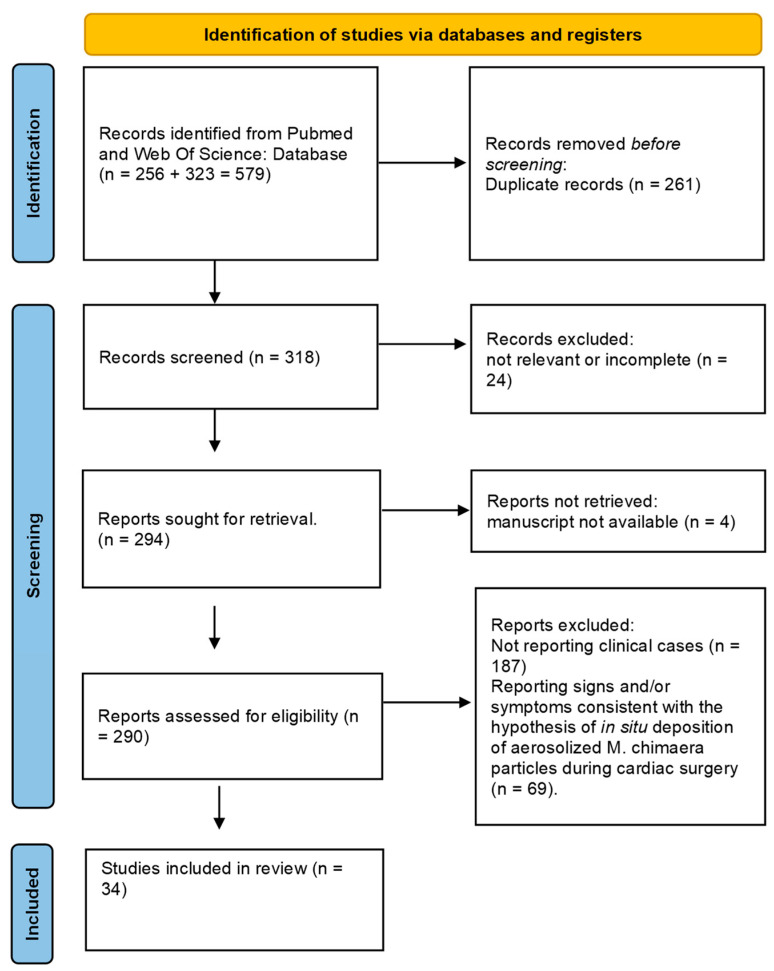
PRISMA 2020 flow diagram for systematic reviews which included searches of databases and registers only.

**Table 1 healthcare-12-01788-t001:** List of references not or improperly consistent with in situ deposition of aerosolised *M. chimaera* particles during surgery.

References	Author, Year	N. of Patients ^§^	Surgery	Mean Time of Presentation If Previous Surgery	Setting (Country)	Organ and/or Tissue Involved
[26]	(Bills et al., 2009)	1	None	Na	Not healthcare (USA)	Lung, nodules in chronic obstructive pulmonary disease
[27]	(Cohen-Bacrie et al., 2011)	1	None	Na	Possible frequent healthcare contact (Réunion Island, FR)	Lung infections in cystic fibrosis
[28]	(Alhanna et al., 2012)	1	None	Na	Not healthcare (Germany)	Lung infection
[29]	(Gunaydin et al., 2013)	5 (of 90)	None	Na	Possible healthcare contact (Turkey)	Lung (reassessment of sputum specimens)
[30]	(Boyle et al., 2015)	125 (of 448)	None	Na	Possible healthcare contact (USA)	Lung (reassessment of sputum specimens)
[31]	(Mwikuma et al., 2015)	1 (of 54)	None	Na	Not healthcare (Zambia)	Lung (reassessment of sputum specimens)
[32]	(Moon et al., 2016)	11	None	Na	Not healthcare (South Korea)	Lung infection (reassessment of sputum specimens)
[33]	(Moutsoglou et al., 2017)	1	None	Na	Not healthcare (USA)	Disseminated with spinal osteomyelitis and discitis
[34]	(Bursle et al., 2017)	1	Tricuspid valve repair and mitral annuloplasty	13 months	Underwent surgery (Australia)	Disseminated
[35]	Kim et al., 2017	8 (of 91)	None	Na	Possible healthcare contact (Korea)	Lung (reassessment of sputum specimens)
[36]	(Chand et al., 2017) *	4	Valvular cardiac surgery	1.15 (0.25–5.1) years	Underwent surgery (UK)	1 osteomyelitis and 3 disseminated
[37]	(Truden et al., 2018)	49 (of 102)	None	Na	Possible healthcare contact (Slovenia)	Lung (reassessment of sputum specimens)
[38]	(Larcher et al., 2019)	4	None	Na	Possible frequent healthcare contact (France)	Lung (reassessment of sputum specimens in cystic fibrosis)
[39]	(Shafizadeh et al., 2019) *	5	Valvular cardiac surgery	20.6 (14–29) months	Underwent surgery (USA)	Disseminated with liver infection
[40]	(Rosero and Shams, 2019)	1	None but operating room nurse 10 years ago	>10 years	Possible frequent healthcare contact (USA)	Lung infection
[41]	(Watanabe et al., 2020)	1	None	Na	Not healthcare (Japan)	Tendons, hand tenosynovitis
[42]	(Chen et al., 2020)	28	None	Na	Not healthcare (Taiwan)	Lung infection (reassessment of sputum specimens)
[43]	(Maalouly et al., 2020)	1	Kidney transplantation	One week	Underwent surgery (Belgium)	Kidney, urinary tract infection in a kidney transplant recipient with concomitant Mycobacterium malmoense lung infection and fibro anthracosis
[44]	(de Melo Carvalho et al., 2020)	1	None	Na	Possible healthcare contact (Portugal)	Disseminated in B-cell lymphoma
[45]	(Sharma et al., 2020)	2	None	Na	Not healthcare (India)	Meninges, meningitis
[23]	(Zabost et al., 2021)	88 (of 200)	None	Na	Possible healthcare contact (Poland)	Lung infection (reassessment of sputum specimens)
[46]	(Kim et al., 2021)	4 (of 320)	None	Na	Possible healthcare contact (Korea)	Lung infection (reassessment of sputum specimens)
[47]	(Kavvalou et al., 2022)	1	None	Na	Possible healthcare contact (Germany)	Central venous catheter infection in cystic fibrosis
[48]	(Robinson et al., 2022)	1	None	Na	Not healthcare (USA)	Lung infection in drug abuser
[49]	(Ahmad et al., 2022)	1	None	Na	Not healthcare (USA)	Lung infection in sarcoidosis
[50]	(George et al., 2022)	1	None	Na	Not healthcare (India)	Skin, periapical abscess with chin ulcer
[51]	(Lin et al., 2022)	1	None	Na	Possible frequent healthcare contact (Taiwan)	Disseminated in adult-onset immunodeficiency syndrome
[52]	(Łyżwa et al., 2022)	1	None	Na	Not healthcare (Poland)	Lung infection in silicosis
[53]	(McLaughlin et al., 2022)	1	Coronary artery bypass grafting	1 year	Underwent surgery (USA)	Tendons, hand tenosynovitis in ipsilateral elbow wound in fisherman
[54]	(Gross et al., 2023)	23	None	Na	Healthcare (USA)	Lung infections in cystic fibrosis (genomic analysis for cluster correlation to hospital outbreaks)
[55]	(Azzarà et al., 2023)	1	None	Na	Possible healthcare contact (Italy)	Lung infection in lung adenocarcinoma treated with immune checkpoint inhibitors
[56]	(Pradhan et al., 2023)	1	Bioprosthetic mitral valve replacement	7 years	Underwent surgery (Australia)	Spinal osteomyelitis and discitis
[57]	(Garcia-Prieto et al., 2024)	1	None	Na	Not healthcare (Spain)	Lung infection in fibro anthracosis
[58]	(Paul et al., 2024)	1	None	Na	Possible healthcare contact (UK)	Lung infection in unilateral pulmonary artery agenesis on the right side

Legend: § in case of respiratory specimens’ reanalysis, the number reflects only those positive for *M. chimaera* (on the total one re-analysed); * The article reports also other cases with cardiac manifestations after cardiac surgery. Na: not applicable.

## Data Availability

No new data were created or analysed in this study. Data sharing is not applicable to this article.

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
