# Peer review of "The State-of-the-Art of Mycobacterium chimaera Infections and the Causal Link with Health Settings: A Systematic Review"

_healthcare, 2024, doi:10.3390/healthcare12171788_

Round 1

Reviewer 1 Report

Comments and Suggestions for Authors

This article is well written; however, the author should revise it due to numerous instances of scarce information in the Materials and Methods section. I believe that only the Discussion section is well organized, and the rest of the manuscript should be revised. I propose a major revision or even rejection, with the possibility of resubmission if major changes are made.

Specific comments:

Line 81: Support this claim with a reference.

Materials and Methods: The Materials and Methods section is completely weak and needs major changes. Include all the relevant sections that a systematic review needs to have. Include PRISMA guidelines and other relevant guidelines such as the Cochrane Handbook for Systematic Reviews of Interventions.

Table 1: Should be more detailed and include more information from the identified studies.

Lines 475-476: This section needs to include all the study’s limitations. Please revise and mention every limitation.

Author Response

Dear Reviewer,

many thanks for your suggestions and considerations on our manuscript that allow us to improve the manuscript and its scientific soundness and relevance. We have answered to single issues (in bold) per points (in italics).

This article is well written; however, the author should revise it due to numerous instances of scarce information in the Materials and Methods section. I believe that only the Discussion section is well organized, and the rest of the manuscript should be revised. I propose a major revision or even rejection, with the possibility of resubmission if major changes are made.

We thank the reviewer. We have revised materials and methods, adding more information on methodology followed for this systematic review. Prisma statement was better explained, as Prisma checklist adoption. 

Specific comments:

Line 81: Support this claim with a reference. We have added Vendramin et al. citation.

Materials and Methods: The Materials and Methods section is completely weak and needs major changes. Include all the relevant sections that a systematic review needs to have. Include PRISMA guidelines and other relevant guidelines such as the Cochrane Handbook for Systematic Reviews of Interventions.

We have revised materials and methods, adding more information on methodology followed for this systematic review. Prisma statement was better explained, as Prisma checklist adoption. 

Table 1: Should be more detailed and include more information from the identified studies.

Many thanks. We have added data on latency in case of surgery, patient/case setting to better correlate to health setting, Country and clear diagnosis. Age and sex, and all comorbidities were avoided as not relevant and consistent to the topic (not a clinical perspective). We have also integrated the text in the Methods and specified better in the Results about respiratory specimens’ reassessment in NMT infection attributed to M. chimaera.

Lines 475-476: This section needs to include all the study’s limitations. Please revise and mention every limitation.

Many thanks. We have described all possible biases and methods used by the authors during research and analysis to reduce their relevance.

Sincerely, vb

Reviewer 2 Report

Comments and Suggestions for Authors

Dear editor,

My opinions about the article titled " The state of art on Mycobacterium chimaera infection and the causal link with health setting: a systematic review " are given below:

In recent years, the importance of Mycobacterium chimaera, a non-tuberculous mycobacterium that is a member of the Mycobacterium avium complex, has increased. It is a bacterium that usually causes infections in the respiratory tract and after cardiothoracic surgery. This rare opportunistic pathogen necessitates careful monitoring due to its potential to cause life-threatening serious diseases and the lengthy identification process. These bacterial characteristics make this systematic review, which brings together research data on a specific topic, valuable.

Basic evaluation of the article:

-        Structured "Abstract" represents the article.

-        In the “Introduction” section, the rationale for the study and the necessity of the study are stated, literature information is given about the difficulties in diagnosis and treatment of Mycobacterium chimaera infections, and the purpose of the study is clearly explained in the last paragraph.

-        The single paragraph "Materials and Methods" section clearly stated the study plan, research materials and procedures. The fact that irrelevant articles that did not contain any data about M. chimae were excluded in the systematic review made us believe that a reliable study was conducted.

-        The article includes a diagram showing the identification of studies through databases and records, and a table describing the consistency of in situ deposition of aerosolized M. chimaera particles during surgery. The figure and table in the "Results" section are essential elements that provide information summarizing the content of the study.

-        In the “Discussion” section, the findings of the study are given under eight subheadings, the limitations are given and the results are evaluated.

-        In the "Conclusions" section, a brief evaluation was made in line with the findings obtained from the study and suggestions were made depending on the results obtained.

-        All references used in the article are relevant. The authors cited only one of their own articles out of 124 references they used (Int. J. Risk Saf. Med. 2023, 34, 129–134). There is no inappropriateness in the use of this auto-citation.

-        The unpublished material file solely includes the "PRISMA 2020 Checklist."

In conclusion, it is appropriate to publish the article in its current form in the journal. However, it would be appropriate to take into consideration the following:

The abbreviation Mc for M. chimaera is not in accordance with bacterial nomenclature. I suggest writing this abbreviation as M. chimaera in italics and making this change consistent throughout the text.

Best regards

Author Response

Dear Reviewer,

many thanks for your suggestions and considerations on our manuscript that allow us to improve the manuscript and its scientific soundness and relevance. We have answered  to single issues (in bold) per points.

My opinions about the article titled " The state of art on Mycobacterium chimaera infection and the causal link with health setting: a systematic review " are given below:

In recent years, the importance of Mycobacterium chimaera, a non-tuberculous mycobacterium that is a member of the Mycobacterium avium complex, has increased. It is a bacterium that usually causes infections in the respiratory tract and after cardiothoracic surgery. This rare opportunistic pathogen necessitates careful monitoring due to its potential to cause life-threatening serious diseases and the lengthy identification process. These bacterial characteristics make this systematic review, which brings together research data on a specific topic, valuable.

Basic evaluation of the article:

-        Structured "Abstract" represents the article.

-        In the “Introduction” section, the rationale for the study and the necessity of the study are stated, literature information is given about the difficulties in diagnosis and treatment of Mycobacterium chimaera infections, and the purpose of the study is clearly explained in the last paragraph.

-        The single paragraph "Materials and Methods" section clearly stated the study plan, research materials and procedures. The fact that irrelevant articles that did not contain any data about M. chimae were excluded in the systematic review made us believe that a reliable study was conducted.

-        The article includes a diagram showing the identification of studies through databases and records, and a table describing the consistency of in situ deposition of aerosolized M. chimaera particles during surgery. The figure and table in the "Results" section are essential elements that provide information summarizing the content of the study.

-        In the “Discussion” section, the findings of the study are given under eight subheadings, the limitations are given and the results are evaluated.

-        In the "Conclusions" section, a brief evaluation was made in line with the findings obtained from the study and suggestions were made depending on the results obtained.

-        All references used in the article are relevant. The authors cited only one of their own articles out of 124 references they used (Int. J. Risk Saf. Med. 2023, 34, 129–134). There is no inappropriateness in the use of this auto-citation.

-        The unpublished material file solely includes the "PRISMA 2020 Checklist."

In conclusion, it is appropriate to publish the article in its current form in the journal. However, it would be appropriate to take into consideration the following:

The abbreviation Mc for M. chimaera is not in accordance with bacterial nomenclature. I suggest writing this abbreviation as M. chimaera in italics and making this change consistent throughout the text.

Best regards

Many thanks for your comments. We have modified the acronym Mc with the more proper M. chimaera as suggested. The table 1 and the text was revised and integrated according to other reviewers' suggestion, such as materials and methods section.

Sincerely vb

Reviewer 3 Report

Comments and Suggestions for Authors

Please check the attached feedback~

Thanks!

Comments on the Quality of English Language

Some sentences are too long, going to several lines, which is, somehow, not easy to read or missing some key points.

Author Response

Dear Reviewer,

many thanks for your suggestions and considerations on our manuscript that allow us to improve the manuscript and its scientific soundness and relevance. We have answered  to single issues (in bold) per points (in italics).

In this manuscript, authors pointed out that it is necessary to study the relationship between
Mycobacterium chimaera (Mc) and healthcare facilities. By reviewing and screening the scientific reports from database, the specific resources were selected for the discussion in depth. In general, Mc characteristics and ecosystem were introduced. Then, contaminations from water system to medical devices, and also implants, were fully stated regarding Mc transmission. Moreover, a critical factor about incubation period linking to symptoms presentation was mentioned for a consideration of pursuing a reasonable timeframe for pathogens research. The lung system related to Mc infections was also included for the purpose of this study. Furthermore, the route of transmitting Mc was stated in detail, mainly including aerosolization of the pathogens. Other subsections, such as approaches for detecting Mc and disinfecting Mc, were given to further discuss a challenge to identify Mc. The last part on causal link assessment demonstrated more in-depth experimental research should be expected to present more scientific evidence for exploring the relationship with healthcare setting and definition of healthcare associated infections. The manuscript is good to me.

Many thanks

Some minor concerns should be considered as shown below:
1. Could the references listed in Table 1 be in order of either “Year” or “Organ and or tissue
involved” for better distinguishing with each other, helping readers to pick the key point?

We have revised the table according to your and the other reviewers' suggestion.

  1. There is confusion on the information from Figure 1 and the main texts from Ln 112 to Ln 123. For instance, the step of “Screening” indicates the records excluded is n=6, but the main texts in Ln 112 mentioned “7 articles”, which means n=7?. Could authors re-edit the information as shown in Figure 1 based on the statement mentioned from Ln 112 to Ln 123? Either way.

Many many thanks. We have made an error during overall manuscript revision, as 1 article was moved from one point to another in Prisma flow chart for coherence. We have revised the flowchart and the text. In fact, 2 articles were excluded in the first step before screening, while 7 (6 not relevant or incomplete, and 1 not available) during screening phase.

  1. As for Mc (4) from Ln 387 and Mc (2) in Ln 388, what is the meaning for that?

Many thanks. We have revised as we have insert wrong references using references processing software. 

Formatting:
a. Ln 4-5, the number followed by each author name should be superscripted

We have modified the text

  1. “PRISMA” in Ln 26, Ln 95, and Ln 109 should be consistently described.

We have revised the text in the methods section and the abstract, also according to another reviewer suggestion. 

  1. The caption of Figure 1 is missing. In Figure 1, the texts should be consistently aligned by one
    style, either “align left” or “center”.

We have revised the text in Figure 1 and added adequate caption.

  1. Ln 47, the period symbol is missing before “The issue of”.

We have modified the text.

  1. Please check if the texts in columns of Table 1 are “capitalized” of first word. and f. Could the title in first row of Table 1 be rotated to “Horizontal”, instead of “Vertical” shown in
    current Table 1?

We have modified the Table also according to another reviewer suggestion. Further data were added. 

  1. The main texts from Ln 135 to Ln 152, please choose the consistent description, such as using
    “2” or “two”, “3” or “three”.

We have revised the text. We have maintained the number in letter if put at the beginning of the sentence.

  1. “MALDI-ToF MS” In Ln 392, Ln 395, and Ln 397 should be consistent.

We have revised the text with MALDI-ToF MS

  1. Corrections: ✓ Ln 161, “6/8 weeks” to “6-8 weeks”, like the texts as shown in Ln 378.
    ✓ Ln 324, “in Trauth and Shaeffer” to “Trauth and Shaeffer”?

We have revised the text.

Some sentences are too long, going to several lines, which is, somehow, not easy to read or missing some key points.

We have overall revised the text, mainly the introduction and the discussion section to improve the text.

Many thanks for the useful considerations.

Sincerely, vb

Round 2

Reviewer 1 Report

Comments and Suggestions for Authors

The author has made a commendable effort in compiling this work. However, there are several points that need to be addressed to enhance the overall quality and rigor of the review:

Line 100: The statement that "PubMed is a database" is incorrect (Medline is the database). Please revise this to accurately reflect PubMed's role as a search engine, rather than a database.

Lines 100-102: To meet the standards of a systematic review, it is recommended that at least two databases be searched (https://www.sciencedirect.com/science/article/pii/S2213398423002725). Some sources suggest that up to four databases should be included to ensure comprehensive coverage (https://systematicreviewsjournal.biomedcentral.com/articles/10.1186/s13643-017-0644-y). Please revise accordingly.

Line 101: The full search algorithm, including the specific search terms and strategies used, should be provided to allow for reproducibility and transparency.

PRISMA Checklist: It is essential to include a PRISMA checklist to ensure that all relevant aspects of the systematic review process have been addressed.

Author Response

Dear Reviewer,

we are glad to answer to your comments (in bold).

In the manuscript, changes are tracked with Word track processor (the text added resulted with different colours, while the text removed is coloured and crossed out), while a clear pdf version of the manuscript is provided.

The author has made a commendable effort in compiling this work. However, there are several points that need to be addressed to enhance the overall quality and rigor of the review:

We particularly appreciated you interest in our article. In fact, it stimulated us in an important work to improve the quality of our review. 

 Line 100: The statement that "PubMed is a database" is incorrect (Medline is the database). Please revise this to accurately reflect PubMed's role as a search engine, rather than a database.

Lines 100-102: To meet the standards of a systematic review, it is recommended that at least two databases be searched (https://www.sciencedirect.com/science/article/pii/S2213398423002725). Some sources suggest that up to four databases should be included to ensure comprehensive coverage (https://systematicreviewsjournal.biomedcentral.com/articles/10.1186/s13643-017-0644-y). Please revise accordingly.

Line 101: The full search algorithm, including the specific search terms and strategies used, should be provided to allow for reproducibility and transparency.

PRISMA Checklist: It is essential to include a PRISMA checklist to ensure that all relevant aspects of the systematic review process have been addressed.

We have modified the text referring to Pubmed search tool and adding Web of science database (evidently setting a data limit at 05,07,2024 but not reported in the query in the text). We have then added research queries and upload as supplementary material the Prisma checklist (that was probably in unpublished material). Abstract and references were revised accordingly.

Many thanks

Sincerely

vb